# The impact of tumor microenvironment and treatment schedule on the effectiveness of radiation therapy

Kyprianos Dimou[1,2], Yiannis Roussakis[2], Constantinos Zamboglou[3,4,5,6], Triantafyllos Stylianopoulos[1]*

1 Cancer Biophysics Laboratory, Department of Mechanical and Manufacturing Engineering, University of Cyprus, Nicosia, Cyprus, 2 Department of Medical Physics, German Medical Institute, Limassol, Cyprus, 3 Department of Radiation Oncology, German Medical Institute, Limassol, Cyprus, 4 Department of Radiation Oncology, University of Freiburg - Medical Center, Freiburg, Germany, 5 German Cancer Consortium (DKTK), Partner Site Freiburg, Freiburg, Germany, 6 School of Medicine, European University Cyprus, Nicosia, Cyprus

* tstylian@ucy.ac.cy

## Abstract

External Beam Radiation Therapy (EBRT) is predominantly administered using Conventionally Fractionated Radiotherapy (CFRT), that is 2 Gy per fraction. However, Moderately Hypofractionated Radiotherapy (MHRT) (approx. 2.5–3 Gy per fraction) and Stereotactic Body Radiotherapy (SBRT) (approx. 6–24 Gy per fraction) regimen are currently clinically investigated or even recently included in standard clinical practice. In addition, hyperfractionated radiotherapy (<1.8–2 Gy per fraction) is also clinically investigated or already used in standard clinical practices. The therapeutic effects of each of these radiotherapy schedules might depend on the degree of radioresistance of the tumor but also on properties of the tumor microenvironment, such as tumor perfusion and oxygenation. Here, building on previous work, we developed a mathematical model to investigate optimal radiotherapy treatment protocols in solid tumors. The model incorporates direct effects of radiation on cancer cells and accounts for the impact of tumor perfusion and oxygenation on the efficacy of radiation therapy. The model was able to accurately reproduce both preclinical and clinical data from different radiotherapy treatment schedules. It confirmed that greater tumor perfusion and thus, oxygenation improves treatment effectiveness by increasing the number of cancer cells killed during the treatment period. It further predicted that this effect is more pronounced for radioresistant tumors, meaning that changes in tumor perfusion of more radioresistant tumors have a greater impact on the percentage of surviving cells at the end of the treatment. The mathematical model provides mechanistic insights into the effectiveness of various radiotherapy schedules and guidelines for how modifying the tumor microenvironment to restore perfusion can affect radiation therapy.

**Data availability statement:** All relevant data are within the paper and its Supporting Information files. The full COMSOL Multiphysics model report is included in the Supporting Information. It contains comprehensive simulation details necessary for replication or further analysis.

**Funding:** Triantafyllos Stylianopoulos European Research Council (ERC) under the European Union's Horizon 2020 research and innovation programme (grant agreement Nos. 863955 and 101141357) Constantinos Zamboglou EU within the framework of the Cohesion Policy Programme "THALIA 2021-2027 The funders had no role in study design, data collection and analysis, decision to publish, or preparation of the manuscript.

**Competing interests:** The authors have declared that no competing interests exist.

## Introduction

Cancer remains a major global health challenge, with an estimated 19.3 million new cases and nearly 10 million deaths worldwide [1]. The most common cancers treatable with radiotherapy – such as breast, head and neck, lung, colorectal and prostate cancers – account for a substantial proportion of these cases [1,2].

Radiation therapy involves targeted delivery of ionizing radiation to kill cancer cells while aiming to limit damage on normal tissues. It is commonly administered as the sole treatment with curative or palliative intent, while also often employed as part of a multimodality regimen. For instance, in many localized breast cancer cases, the tumor is removed surgically and radiotherapy is applied to increase local control while hormonal therapies or chemotherapy are utilized to eliminate microscopic local and distant metastasis [3]. The main mechanism behind the therapeutic effect of radiation is DNA damage. Specifically, radiation induces breaks in the DNA. When essential regions of the DNA are damaged, cancer cells are unable to replicate and undergo apoptosis, while normal cells possess efficient DNA repair mechanisms, which can fix the majority of such DNA damage [4].

Traditionally, curative-intent External Beam Radiation Therapy (EBRT) is predominantly administered using Conventionally Fractionated Radiotherapy (CFRT), that is 1.8–2 Gy per fraction [5]. However, some cancers exhibit favorable or similar sensitivity when irradiated with higher dose per fraction, and fewer total fractions (e.g., prostate) [6]. To increase treatment efficiency and to enhance patients' convenience, Moderately Hypofractionated Radiotherapy (MHRT: approx. 2.5–3 Gy per fraction) and Stereotactic Body Radiotherapy (SBRT: approx. 6–24 Gy per fraction) regimen are currently clinically investigated or even recently included in standard clinical practice [7–11]. Particularly, several randomized-controlled trials reported on comparable oncologic outcomes without or with very low increase in toxicity [7–14]. On the other hand, some cancers respond favorably to lower dose per fraction and more total fractions (e.g., some lung or head-and-neck cancer cases). In these cases, hyperfractionated radiotherapy (<1.8–2 Gy per fraction) is clinically investigated or already used in standard clinical practices [15,16].

Mathematical models have helped comprehend tumor growth dynamics as well as tumor response to radiotherapy. Discrete/stochastic models constitute one of the two main categories of mathematical models. This type of models focus on the microscopic scale, represents cells in a discrete way and accounts for interactions at the cellular level [17]. An example of a discrete model which employs radiotherapy deals with tumor growth and response to radiotherapy of patients with glioblastoma multiforme [18]. In addition, this model considers different radiosensitivity values for the cell cycle phases as well as the influence of the genetic profile of the tumor. Continuous models are the other main category of mathematical models developed for radiotherapy, describing events at the macroscopic scale [17]. Such a model has been generated to distinguish the cells in the four cell cycle phases $G_1$, S, $G_2$, M in a continuum fashion, as well as to describe the growth of tumor cells and their response to radiotherapy [19]. To simulate the impact of radiotherapy on an advanced diffusive glioma, a diffusive model has been developed, employing the linear quadratic model

[20]. In addition, this modeling approach has been extended to study the dynamic growth of head-and-neck tumors during radiotherapy [21,22], utilizing clinical data for the validation of the models.

To this end, building on previous work, we developed a continuous mathematical model to investigate optimal radiotherapy treatment protocols in solid tumors. The model considers the radiotherapy schedule, the different radiosensitivity values of each cell phase but also expands to incorporate properties of the tumor microenvironment, such as tumor perfusion and oxygenation, that are known to play an important role in the efficacy of the treatment [23,24]. To our knowledge, this is the first continuous mathematical model to explore different radiotherapy schedules, while taking into account changes in perfusion and oxygenation. Our model is validated based on pre-clinical [25,26] as well as clinical data [16] from the literature and provides insights into the effectiveness of various radiotherapy treatment protocols for prostate as well as head and neck cancers.

## Methods

### Model description

The mathematical model employed in this study integrates the dynamics of the cancer cell cycle, oxygen supply, radiotherapy response, and tumor growth mechanics within a multiphysics framework. Cancer cells are classified into five compartments ($G_1$, S, $G_2$, M, $G_0$ phases), each representing a distinct stage of the cell cycle. Transitions between these phases are governed by a system of coupled ordinary differential equations (Eqs. 1–5), with oxygen concentration modulating the proliferative or quiescent behavior of cells. Radiotherapy-induced cell death is modeled using the Linear Quadratic (LQ) model, incorporating radiosensitivity parameters that vary with oxygen availability. Simultaneously, the spatial and temporal distribution of oxygen concentration is governed by a diffusion-reaction equation (Eq. 10). Tumor expansion is described through volumetric growth using the multiplicative decomposition of the deformation gradient tensor (Eqs. 11–17). This comprehensive approach, solved via COMSOL Multiphysics, enables the simultaneous simulation of cellular kinetics, therapy response, and biomechanical deformation of the tumor microenvironment.

### Cell cycle

The cell cycle of cancer cells is distinguished into the mitotic phase (M-phase), the quiescent phase ($G_0$-phase) and the interphase ($G_1$, S, $G_2$ phases) [27]. The dynamics of the five phases are shown in the schematic diagram in Fig 1, in which the transitions among them are depicted. The mass balance of each compartment is given by the equations:

$$\frac{dG_1}{dt} = 2k_M M + k_{G_0 G_1} G_0 - \widetilde{\alpha} k_{G_1} G_1 - \widetilde{b} k_{G_1 G_0} G_1 - \mu_{G_1} G_1 - R_{G_1} G_1 \tag{1}$$

$$\frac{dS}{dt} = \widetilde{\alpha} k_{G_1} G_1 - k_S S - \mu_S S - R_S S \tag{2}$$

$$\frac{dG_2}{dt} = k_S S - k_{G_2} G_2 - \mu_{G_2} G_2 - R_{G_2} G_2 \tag{3}$$

$$\frac{dM}{dt} = k_{G_2} G_2 - k_M M - \mu_M M - R_M M \tag{4}$$

$$\frac{dG_0}{dt} = \widetilde{b} k_{G_1 G_0} G_1 - k_{G_0 G_1} G_0 - \mu_{G_0} G_0 - R_{G_0} G_0 \tag{5}$$

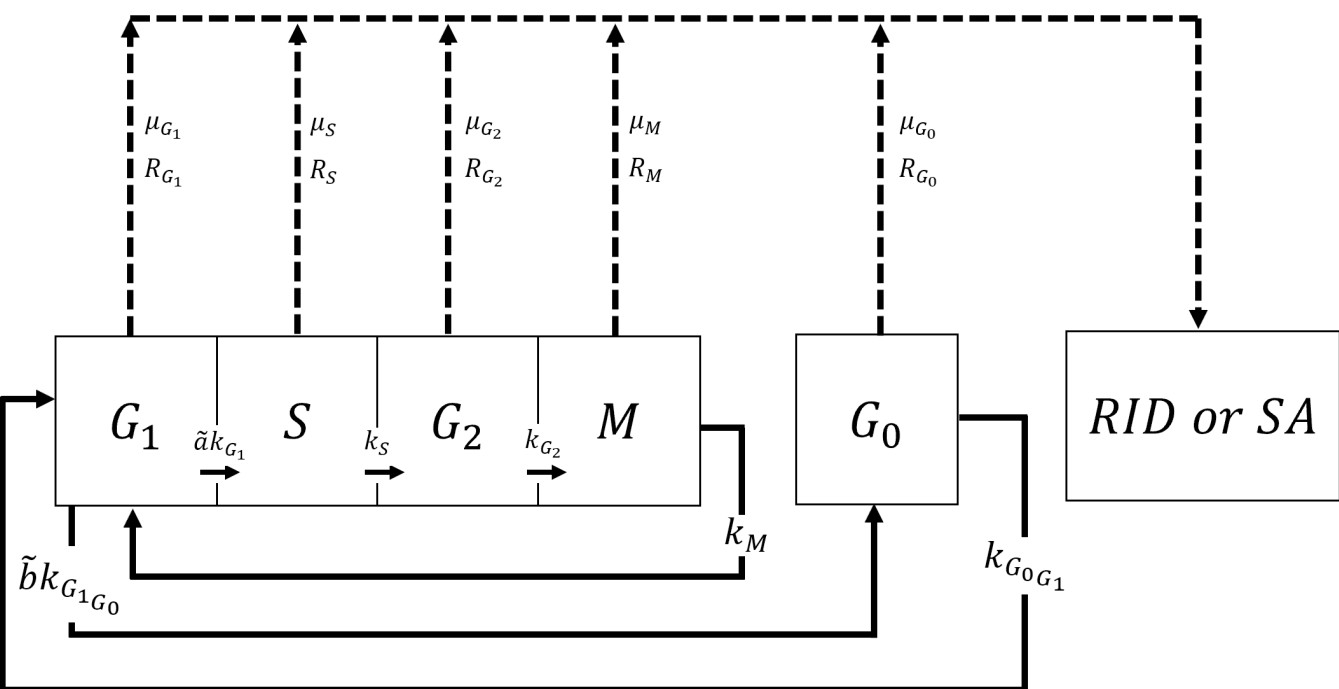

**Fig 1. Cytokinetic model of a tumor cell.** Explanation of symbols: $G_1$: $G_1$ (gap1)-phase, S: DNA synthesis phase, $G_2$: $G_2$ (gap2)-phase, M: Mitosis, $G_0$: Quiescent phase, RID: Radiation-induced death, SA: Spontaneous apoptosis.

Equation (1) describes the cancer cells in the $G_1$-phase (gap1). The first term on the right-hand side represents the transition of the newly divided cancer cells from the M-phase to the $G_1$-phase, at a rate of $k_M$. The coefficient 2 of this term originates from the fact that during cell division, two identical daughter cells are created [19]. The second term in Equation (1) describes the transition of cells from $G_0$-phase to the $G_1$-phase at a rate of $k_{G_0 G_1}$. The rest of the terms in Equation (1) are all sink terms. $\widetilde{\alpha} k_{G_1} G_1$ and $\widetilde{b} k_{G_1 G_0} G_1$ illustrate the cells transitioning from the $G_1$-phase to the S-phase at a rate of $k_{G_1}$ and the transition from $G_1$-phase to the $G_0$-phase at a rate of $k_{G_1 G_0}$, respectively. Terms $\widetilde{\alpha}$ and $\widetilde{b}$ given by Equations (6) and (7) are variables which correlate oxygen concentration with the cellular phase transition:

$$\widetilde{\alpha} = 0.4 \frac{c_{ox}}{c_{iox}} + 0.3 \tag{6}$$

$$\widetilde{b} = 1 - \widetilde{\alpha} \tag{7}$$

where $c_{ox}$ is the oxygen concentration in the tissue and $c_{iox}$ is the initial oxygen concentration. Specifically, elevated oxygen levels lead variable $\widetilde{\alpha}$ to its upper limit, indicating an increased transition of cells from the $G_1$-phase to the S-phase for further division. Conversely, reduced oxygen levels cause an increase in parameter $\widetilde{b}$, resulting in a higher number of cells shifting from the $G_1$-phase to the quiescent $G_0$-phase. In addition, Equations (6) and (7) have been designed in such a way, that the upper and lower limits of parameters $\widetilde{\alpha}$ and $\widetilde{b}$ are 0.7 and 0.3, respectively [28]. Finally, Equation (1) concludes with two terms quantifying cell death via spontaneous apoptosis at a rate of $\mu_{G_1}$ and cell death due to radiotherapy at a rate of $R_{G_1}$.

Equation (2) describes the cancer cells in the S-phase (DNA synthesis). The first term on the right-hand side acts as a source term, representing the influx of cells from the $G_1$-phase. The rest of the terms function as sink terms. $k_S$ denotes

the rate at which cells exit the S-phase and transition to the $G_2$-phase. Once more, the last two terms illustrate cell death via spontaneous apoptosis at a rate of $\mu_S$ and cell death due to radiotherapy at a rate of $R_S$.

Proceeding to Equation (3), which characterizes the cancer cells in the $G_2$-phase (gap2), it features one source term and three sink terms, similar to Equation (2). $k_S S$ is the source term derived from the S-phase. Moreover, cells shift from the $G_2$-phase to the M-phase at a rate of $k_{G_2}$ and die due to spontaneous apoptosis at a rate $\mu_{G_2}$ and due to radiotherapy at a rate $R_{G_2}$.

Equation (4) illustrates the cancer cells in the M-phase (Mitosis). The source term on the right-hand side quantifies the cells leaving the $G_2$-phase and entering the M-phase at a rate of $k_{G_2}$. $k_M M$ functions as a sink term, representing the cells transitioning from the M-phase back to the $G_1$-phase. Additionally, the final two terms account for cell death: spontaneous apoptosis at a rate of $\mu_M$ and radiation-induced cell death at a rate of $R_M$.

Finally, Equation (5) describes the cancer cells in the $G_0$-phase (quiescent). Cells enter the $G_0$-phase from the $G_1$-phase at a rate of $\widetilde{b} k_{G_1 G_0}$ and exit the $G_0$-phase to proceed to the $G_1$-phase at a rate of $k_{G_0 G_1}$. In the same manner as the other equations, cells die due to spontaneous apoptosis at a rate $\mu_{G_0}$ and due to radiotherapy at a rate $R_{G_0}$.

The cancer cell densities $G_1, S, G_2, M, G_0$ were normalized by division with a reference initial value of $10^7$ cells/cm$^3$ [29]. Therefore, the initial value of the sum of all five phases was set to 1 for the tumor region and to 0 for the host tissue. Particularly, cells were assumed to be in the $G_1$-phase at the start of every simulation [27].

A summary of the model parameter descriptions and units is given in Table 1.

## Radiotherapy

Radiotherapy-induced cell death is calculated based on the Linear Quadratic (LQ) Model, which is extensively used in the relevant literature [18,28,41–44]. The death rate of cells after exposure to a uniform radiation dose $D$ is given by:

$$R_C = \begin{cases} 0 & \text{for } t \neq \text{therapy} \\ (1 - S_{f_c})/tau & \text{for } t = \text{therapy} \end{cases}$$

$$S_{f_c} = e^{-\alpha_c D - \beta_c D^2}$$

$$c = G_1, S, G_2, M, G_0 \tag{8}$$

where $S_{f_c}$ represents the survival fraction of each phase ($c = G_1, S, G_2, M, G_0$) and $\alpha$ (Gy$^{-1}$) and $\beta$ (Gy$^{-2}$) describe the initial slope and curvature, respectively, of the survival curve and are associated with cell radiosensitivity [18]. $tau$ represents the duration of the radiotherapy session according to clinical practice and is equal to 6.5 min. It is known that cell radiosensitivity varies significantly throughout the cell cycle. For instance, the S-phase is considered to be the most resistant one. Therefore, each cell phase is modelled with different $\alpha$ and $\beta$, resulting in different death rates after exposure to radiation [18,45]. A summary of the values of $\alpha$ and $\beta$ used in this study is presented in Tables 2–4 for head and neck and prostate tumors.

## Dead cancer cells

After every radiotherapy treatment fraction, a number of dead cancer cells is produced:

$$\frac{dN_{tot}}{dt} = \sum_{n=1}^{Q} \frac{dN_n}{dt}$$

**Table 1. Model parameter descriptions and units.**

| Parameter | Description | Value | Reference |
|---|---|---|---|
| $k_M$ | Transition rate of the newly divided cancer cells from the $M$-phase to the $G_1$-phase | Determined by data fitting | This work |
| $k_{G_0 G_1}$ | Transition rate of cells from the $G_0$-phase to the $G_1$-phase | 0.024 [1/day] | [30] |
| $k_{G_1}$ | Transition rate of cells from the $G_1$-phase to the $S$-phase | 0.035 [1/hour] | [31] |
| $k_{G_1 G_0}$ | Transition rate of cells from the $G_1$-phase to the $G_0$-phase | 0.09 [1/day] | [27] |
| $k_S$ | Transition rate of cells from the $S$-phase to the $G_2$-phase to the | 0.67 [1/hour] | [32] |
| $k_{G_2}$ | Transition rate of cells from the $G_2$-phase to the $M$-phase | 0.847 [1/day] | [33] |
| $\mu_{G_1}$ | Spontaneous apoptosis death rate of $G_1$-phase | 0.01 [1/day] | [34] |
| $\mu_S$ | Spontaneous apoptosis death rate of $S$-phase | 0.01 [1/day] | [34] |
| $\mu_{G_2}$ | Spontaneous apoptosis death rate of $G_2$-phase | 0.01 [1/day] | [34] |
| $\mu_M$ | Spontaneous apoptosis death rate of $M$-phase | 0.01 [1/day] | [34] |
| $\mu_{G_0}$ | Spontaneous apoptosis death rate of $G_0$-phase | 0.01 [1/day] | [34] |
| $\tilde{\alpha}$ | Oxygen variable related to the transition of cells from $G_1$ to $S$-phase | 0.3 - 0.7 | [28] |
| $\tilde{b}$ | Oxygen variable related to the transition of cells from $G_1$ to $G_0$-phase | 0.7 - 0.3 | [28] |
| $D_{ox}$ | diffusion coefficient of oxygen | $1.55 \times 10^{-4}$ m$^2$ day$^{-1}$ | [35] |
| $P_{er}$ | vascular permeability of oxygen | $3.55 \times 10^{-4}$ m s$^{-1}$ | [36] |
| $S_v$ | functional vascular density | 7000 m$^{-1}$ | [37] |
| $c_{iox}$ | oxygen concentration in the vessels, initial oxygen concentration | 0.2 mol m$^{-3}$ | [38] |
| $A_{ox}$ | oxygen uptake parameter | 2200 mol m$^{-3}$ day$^{-1}$ | [35,38] |
| $k_{ox}$ | oxygen uptake parameter | 0.00464 mol m$^{-3}$ | [35,38] |
| $\mu$ | Shear modulus | 5 kPa for host tissue; 10.4 kPa for tumor | [39,40] |
| $k$ | Bulk modulus | 6.67 kPa for host tissue; $10.40 \times 10^7$ kPa for tumor | [39,40] |

$$\frac{dN_n}{dt} = R_{G_{1n}}G_1 + R_{S_n}S + R_{G_{2n}}G_2 + R_{M_n}M + R_{G_{0n}}G_0 \tag{9}$$

where $n$ and $Q$ denote the session number and the total number of sessions for each radiotherapy schedule, respectively.

**Oxygen concentration**

The rate of change of oxygen concentration in the tumor and surrounding host tissue is given by a diffusion-reaction type equation [17,41,35,36,46]. The reaction term is related to the oxygen transferred from the vessels to the tumor minus the amount of oxygen consumed by cells:

$$\frac{\partial c_{ox}}{\partial t} = D_{ox}\nabla^2 c_{ox} + P_{er}S_v(c_{iox} - c_{ox}) - \frac{A_{ox}c_{ox}}{c_{ox} + k_{ox}}T \tag{10}$$

where $D_{ox}$ is the diffusion coefficient of oxygen, $P_{er}$ is the vascular permeability of oxygen, which characterizes diffusion across the tumor vessel wall, $S_v$ is the functional vascular density, $c_{iox}$ is the oxygen concentration in the vessels and the initial oxygen concentration, $A_{ox}$ and $k_{ox}$ are oxygen uptake parameters and $T$ is the total number of cancer cells, i.e., the summation of all cell cycle phases, normalized by division with a reference initial value of $10^7$ cells/cm$^3$ [29].

## Tumor growth

The growth of an idealized spherical tumor surrounded by normal tissue is based on principles of continuum mechanics. The deformation gradient tensor $\overline{\overline{F}}$ was decomposed into two components [17,36,47,48]:

$$\overline{\overline{F}} = \overline{\overline{F}}_e \overline{\overline{F}}_g \tag{11}$$

where $\overline{\overline{F}}_g$ is the inelastic component, which accounts for volumetric growth that does not generate mechanical stress and $\overline{\overline{F}}_e$ is the elastic component, which accounts for mechanical interactions of the tumor with the surrounding normal tissue. The inelastic component $\overline{\overline{F}}_g$ is assumed to be homogenous and isotropic [35,36,46,49].

$$\overline{\overline{F}}_g = \lambda_g I = \begin{bmatrix} \lambda_g & 0 & 0 \\ 0 & \lambda_g & 0 \\ 0 & 0 & \lambda_g \end{bmatrix} \tag{12}$$

where $\lambda_g$ is the growth stretch ratio, considering cancer cell proliferation and death. The elastic component $\overline{\overline{F}}_e$ is quantified as:

$$\overline{\overline{F}}_e = \overline{\overline{F}} \overline{\overline{F}}_g^{-1} \tag{13}$$

The growth stretch ratio is quantified based on the summation of the mass balances of the cell cycle phases. Particularly, the rate of change of the growth stretch ratio is calculated as [36]:

$$\frac{3}{\lambda_g} \frac{d\lambda_g}{dt} = -\mu_{G_0} G_0 - \mu_{G_1} G_1 - \mu_S S - \mu_{G_2} G_2 - \mu_M M + k_M M - C_r N_{tot} \tag{14}$$

where $C_r$ is the clearance rate which defines the rate at which dead cancer cells are cleared from the tumor.

For a quasistatic problem with no body forces, the Stress Balance is written as [17,36]:

$$\nabla \cdot \overline{\overline{\sigma}} = 0 \tag{15}$$

The Cauchy stress tensor $\overline{\overline{\sigma}}$ is described as [17,36,50]:

$$\overline{\overline{\sigma}}^s = \frac{1}{J_e} \overline{\overline{F}}_e \frac{\partial W}{\partial \overline{\overline{F}}_e^T} \tag{16}$$

where $J_e = \det(\overline{\overline{F}}_e)$ and $W$ is the neo-Hookean strain energy density function [17,36,51]:

$$W = \frac{\mu(-3 + I_1)}{2} + \frac{K(-1 + J_e)^2}{2} \tag{17}$$

where $\mu$ is the shear modulus, $I_1 = tr\overline{\overline{C}}_e$ is the first invariant of the elastic Cauchy-Green deformation tensor $\overline{\overline{C}}_e = \overline{\overline{F}}_e^T \overline{\overline{F}}_e$ and $K$ is the bulk modulus.

The model features a spherical tumor with an initial diameter of 500 µm, positioned at the center of a cubic host domain. In addition, the cubic host domain is two orders of magnitude larger to prevent boundary effects from influencing the tumor's growth [52]. Due to symmetry, only one-eighth of the entire system was analyzed (Fig 2). In addition, symmetry boundary conditions for the displacements $u$ and the oxygen concentration were employed at the symmetry planes:

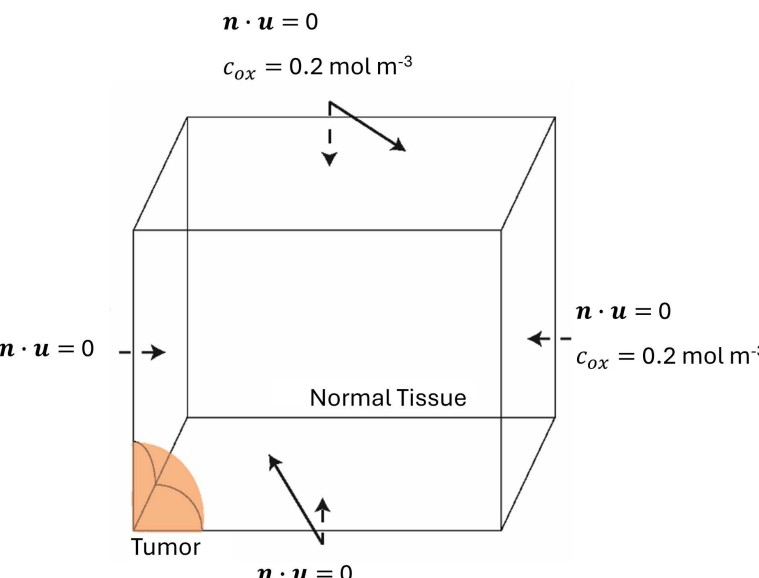

**Fig 2. Geometry of the computational domain and boundary conditions applied.**

$$\overline{n} \cdot \overline{u} = 0 \tag{18}$$

$$c_{ox} = 0.2 \text{ mol m}^{-3} \tag{19}$$

All the above equations were solved concurrently using the commercial finite element software COMSOL Multiphysics (COMSOL, Inc., Burlington, MA, USA). Values for the model parameters are provided in Table 1. Finally, COMSOL automatically applied the continuity conditions for stresses, displacements and oxygen concentrations at the interface with the normal tissue.

### Effect of tumor perfusion/oxygenation on treatment efficacy

Hypoxia refers to the condition where there is inadequate oxygen in the tissue microenvironment. It is a hallmark of solid tumors patho-physiology caused by vascular abnormalities and the high demands on oxygen supply by the rapidly growing cancer cells [53]. In other words, oxygen demands surpass the oxygen supply [54–56]. A hypoxic environment directly promotes the malignant characteristics of cancers, including metastasis and resistance to radiotherapy [54,56,57]. Radiotherapy's ability to kill tumor cells primarily depends on its DNA-damaging effects. This damage can happen either directly or indirectly. Direct damage refers to the interaction of radiation with the DNA molecules and is responsible for 30–40% of the induced damage. Indirect damage denotes the DNA damage caused by reactive oxygen species (ROS), which are formed from the interaction of radiation with water molecules, and accounts for 60–70% of the induced damages [56,58,59]. The subsequent DNA damage is readily reversible unless oxygen is present, in which case it reacts with the damaged DNA and stabilizes the lesion [56,59]. Simply put, oxygen inhibits the repair of DNA damage. As a result, unrepaired DNA damage eventually leads to the death of the cancer cell. Several studies have examined the effects of oxygen on the efficiency of radiotherapy in greater detail [60–62]. Their findings demonstrated that cells were significantly more sensitive to radiation when oxygen was present. Specifically, to quantify the increase in radiation sensitivity of cells due to the presence of oxygen, they used the oxygen enhancement ratio (*OER*). *OER* is the ratio which compares the amount of radiation needed to produce a certain level of biological effect under hypoxic conditions, to the amount needed under normal oxygenated conditions:

$$OER = \frac{Radiation\ dose\ in\ hypoxia}{Radiation\ dose\ in\ normoxia} \tag{20}$$

Crucially, the radiation dose needed to produce the same biological effect is approximately three times greater during hypoxia, compared to when oxygen levels are normal [56,59,62].

Based on the considerations outlined above, we incorporated mechanisms into our model to simulate the effects of oxygen variability on tumor growth. Particularly, right before the first treatment session, the value of the functional vascular density $S_v$ is altered. Low and high vascular densities are represented by values 5000 m$^{-1}$ and 25000 m$^{-1}$ respectively [37]. Consequently, low vascular density leads to reduced perfusion and oxygen levels within the tumor region, while high vascular density is associated with normal perfusion and oxygen levels. Subsequently, $OER$ is calculated based on the levels of oxygenation.

$$OER = -10c_{ox} + 3 \tag{21}$$

Equation 21 was design in such a way that $OER$ is inversely proportional to the concentration of oxygen and follows a linear manner. The upper and lower threshold of $OER$ is 3 and 1 respectively [18,63]. Moving on, $\alpha$ and $\beta$ parameters of the Linear Quadratic model are recalculated:

$$\alpha_{reduced} = \frac{\alpha}{OER}, \quad \beta_{reduced} = \frac{\beta}{OER^2} \tag{22}$$

Finally, the survival fraction based on the Linear Quadratic model is evaluated, using the altered $\alpha$ and $\beta$ parameters.

## Results

### Model validation with preclinical data

Our model was first validated based on two different murine studies from the literature. The one study was conducted on murine HPV-positive head and neck tumors [25]. Specifically, the murine orthotopic head and neck tumor models were treated using local hypofractionated radiotherapy either alone or combined with systemic administration of the FAP-CD40 antibody. For the purposes of our study, only the scenario involving hypofractionated radiotherapy alone was utilised. Tumors received two consecutive 6 Gy doses of radiation on days 10 and 11 post-tumor engraftment. Tumor size was monitored twice weekly using a caliper. To match the simulation model's predictions with the study's results, two parameters were adjusted. Firstly, the transition rate of the newly divided cancer cells from the $M$-phase to the $G_1$-phase, $k_M$, was fitted so that the simulated tumor volume matched the average experimental tumor volume of the six untreated (control) murine cases. Subsequently, the clearance rate $C_r$, which was applied in the context of radiotherapy, was fitted to ensure that the model's predictions aligned with the study's results and matched the radiotherapy schedule. A summary of the values used for the fitting of the model to the data of the murine HPV-positive head and neck tumors study [25] as well as the model predictions are presented in Table 2 and Fig 3, respectively. It is evident that the model shows good agreement with the experimental data, as quantified by the coefficient of determination $R^2$ (Fig 3). In general, an $R^2$ value of 1 indicates a perfect fit between the simulated and experimental data. Accordingly, the transition rate $k_M$ and the clearance rate $C_r$ were adjusted to the control and treated groups respectively, to maximize $R^2$, aiming for values as close to 1 as possible. Additionally, we conducted a sensitivity analysis to evaluate how the transition rate $k_M$, the clearance rate $C_r$ and the parameter $\alpha$ of the Linear Quadratic model influence the model outputs. Further information regarding the analysis and its results are provided in S1 Text, S1 Table and S11-S13 Figs in Supporting Information.

**Table 2. Model parameters and units derived from the fitting of model to the data from murine HPV-positive head and neck tumors [25].**

| Parameter | Value | Reference |
| --- | --- | --- |
| $k_M$ (hour$^{-1}$) | 0.02 | fitted to control group |
| $C_r$ (day$^{-1}$) | 1/15 | fitted to treated group |
| $\frac{\alpha}{\beta}$ (Gy) | 14.3 | [44] |
| $\alpha_{G_1}$ (Gy$^{-1}$) | 0.04 | [28,43] |
| $\alpha_S$ (Gy$^{-1}$) | 0.03 | [28,43] |
| $\alpha_{G_2}$ (Gy$^{-1}$) | 0.04 | [28,43] |
| $\alpha_M$ (Gy$^{-1}$) | 0.04 | [28,43] |
| $\alpha_{G_0}$ (Gy$^{-1}$) | 0.02 | [28,43] |

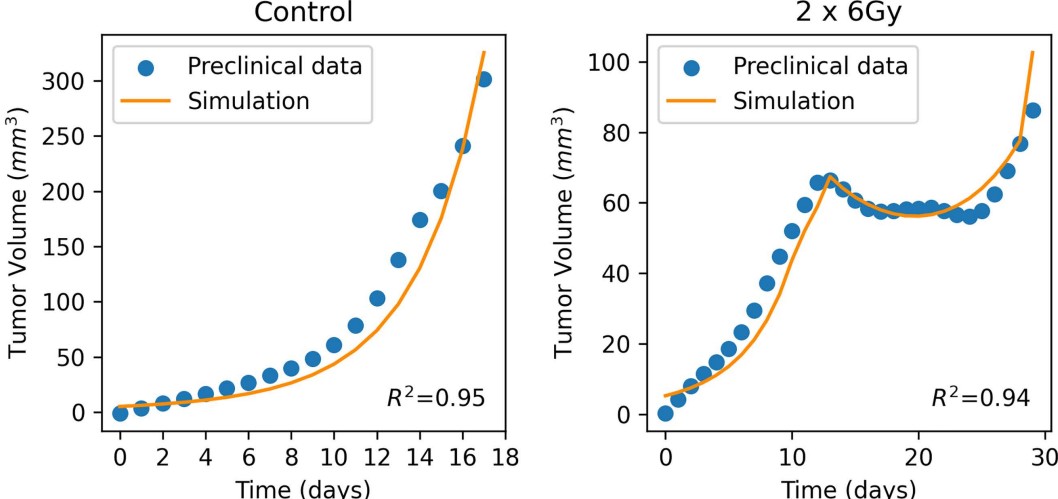

**Fig 3. Validation of the model on murine HPV-positive head and neck tumors.** Dots denote the average value of the experimental data [25] and the solid line the model predictions.

The second study was conducted on murine prostate cancer models [26]. Mice were divided into four treatment groups once the tumors reached 100mm³. Each treatment group received different doses (2 Gy, 5 Gy) and varying numbers of fractions (1 fraction, 3 fractions, 5 fractions). The primary variable studied was tumor growth delay, which was defined as a significant increase in time (days) for a tumor treated at 100 mm³ to reach an end-point size of 400 mm³ compared to control untreated tumors. Similarly to the murine HPV-positive head and neck tumors study, the parameter $k_M$ was adjusted based on the control group. Subsequently, the clearance rate $C_r$ was adjusted on the 3 x 5 Gy case in order to ensure that the model's predictions matched the study's results. Next, the fractionated dose and the number of fractions were modified for the remaining four cases in accordance with the study's protocol. Importantly, all the other model parameters remained unchanged. This approach allowed us to evaluate how accurately the model predicts the outcomes of the remaining treatment protocols. An overview of the values used for the validation of the murine prostate cancer study [26] as well as the model predictions are presented in Table 3 and Fig 4, respectively. Finally, the model demonstrates good alignment with the experimental data, as reflected by the coefficient of determination $R^2$ (Fig 4).

**Table 3. Model parameters and units derived from the fitting of the model to data from murine prostate cancer [26].**

| Parameter | Value | Reference |
|---|---|---|
| $k_M$ (hour$^{-1}$) | 0.0015 | fitted to control group |
| $C_r$ (day$^{-1}$) | 1/17 | fitted to treated group 3 x 5 Gy |
| $\frac{\alpha}{\beta}$ (Gy) | 14.3 | [44] |
| $\alpha_{G_1}$ (Gy$^{-1}$) | 0.01 | [43] |
| $\alpha_S$ (Gy$^{-1}$) | 0.01 | [43] |
| $\alpha_{G_2}$ (Gy$^{-1}$) | 0.01 | [43] |
| $\alpha_M$ (Gy$^{-1}$) | 0.01 | [43] |
| $\alpha_{G_0}$ (Gy$^{-1}$) | 0.01 | [43] |

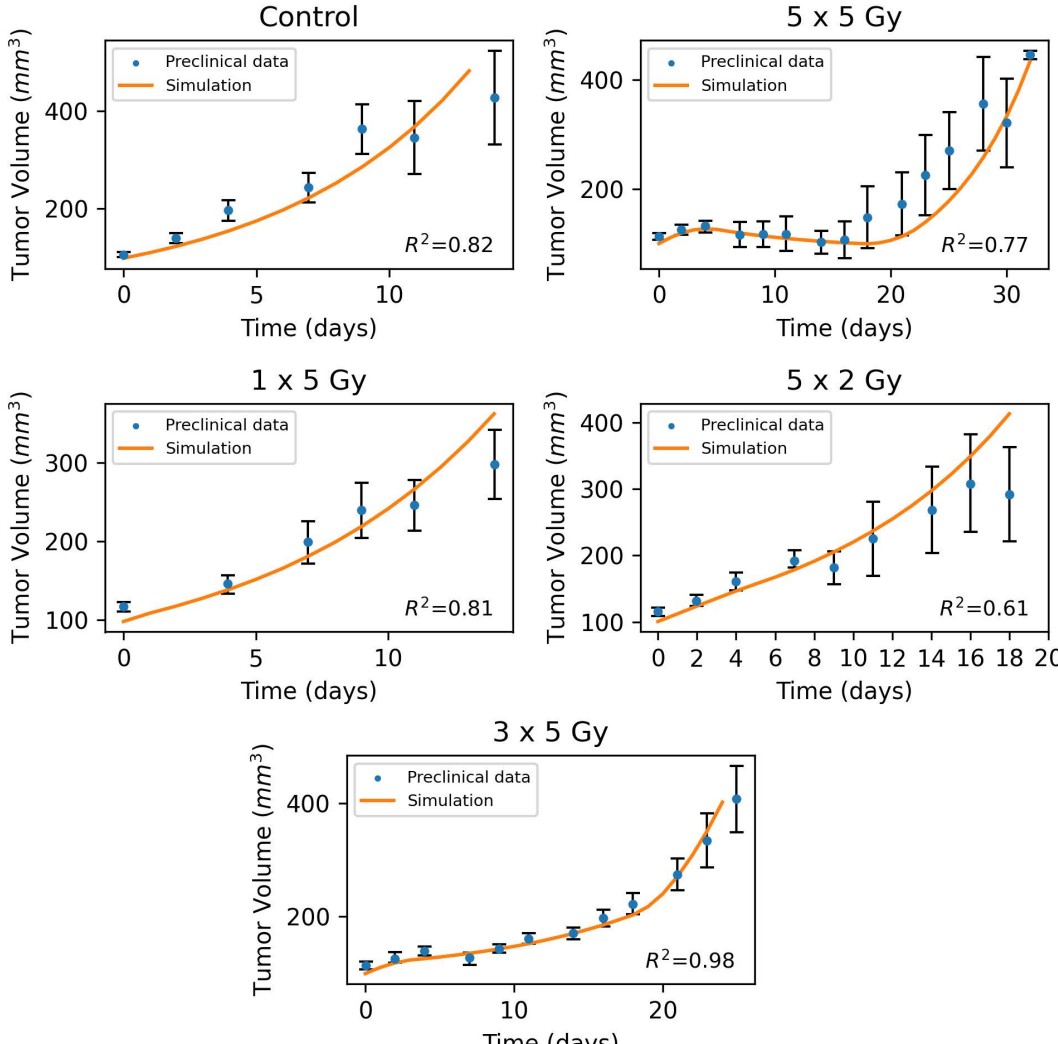

**Fig 4. Validation of the model with data from murine prostate cancer tumors [26].** Dots denote the average value of the experimental data and the solid line the model predictions.

## Model validation with human studies

For further validation of our model, we incorporated data from 10 head-and-neck cancer patients, enrolled at The University of Texas M.D. Anderson Cancer Center [16]. Patients were treated with definitive external beam radiotherapy using either conventional or conformal treatment techniques. Specifically, the clinical data included 5 patients receiving 72 Gy in 42 fractions, 1 patient receiving 60 Gy in 30 fractions and 4 patients receiving 70 Gy in 35 fractions. The study was focused on quantifying the volumetric changes of the tumors during the course of radiotherapy. Similarly to the murine studies, $k_M$ and the clearance rate $C_r$ were adjusted on each one of the treated cases, in order to ensure that the model's predictions align with the study's outcome. An overview of the values used for the validation of the human head-and-neck cancer patients study [16] is presented in Tables 4 and 5. The validation of the model is illustrated in Fig 5 and Fig 6. Once more, the model exhibits good correspondence with the experimental data, as indicated by the coefficient of determination $R^2$ (Fig 5, Fig 6). In the cases with lower $R^2$ values (HN01 and HN02), the clinical data exhibited substantial fluctuations in tumor volume over time, unlike the smoother and more consistent decline observed in the other patients. As a result, the optimal approach for fitting our simulation curve was to capture the overall trend by passing through the general pattern of data points, rather than closely overlapping each point as in the cases with smoother response curves. To complement the goodness-of-fit assessment based on $R^2$ values, S1-S10 Figs in Supporting Information present the Bland-Altman plots, providing an additional perspective on the agreement between the model-predicted and experimental tumor volumes.

**Table 4. Model parameters and units of the fitting of the model to clinical data of head-and-neck cancer patients [16].**

| Parameters | Value | Reference |
|---|---|---|
| $k_M$ (hour$^{-1}$) | 0.008 | This work |
| $\frac{\alpha}{\beta}$ (Gy) | 8 | [45] |
| $\alpha_{G_1}$ (Gy$^{-1}$) | 0.25 | [45] |
| $\alpha_S$ (Gy$^{-1}$) | 0.182 | [45] |
| $\alpha_{G_2}$ (Gy$^{-1}$) | 0.590 | [45] |
| $\alpha_M$ (Gy$^{-1}$) | 0.595 | [45] |
| $\alpha_{G_0}$ (Gy$^{-1}$) | 0.236 | [45] |

**Table 5. Clearance rate derived from the fitting of the model to the clinical data of the head-and-neck cancer patients [16].**

| Patient | $C_r$ (day$^{-1}$) | Reference |
|---|---|---|
| HN01 | 1/325 | This work |
| HN02 | 1/250 | This work |
| HN05 | 1/40 | This work |
| HN06 | 1/70 | This work |
| HN07 | 1/175 | This work |
| HN08 | 1/150 | This work |
| HN09 | 1/250 | This work |
| HN10 | 1/70 | This work |
| HN13 | 1/20 | This work |
| HN15 | 1/175 | This work |

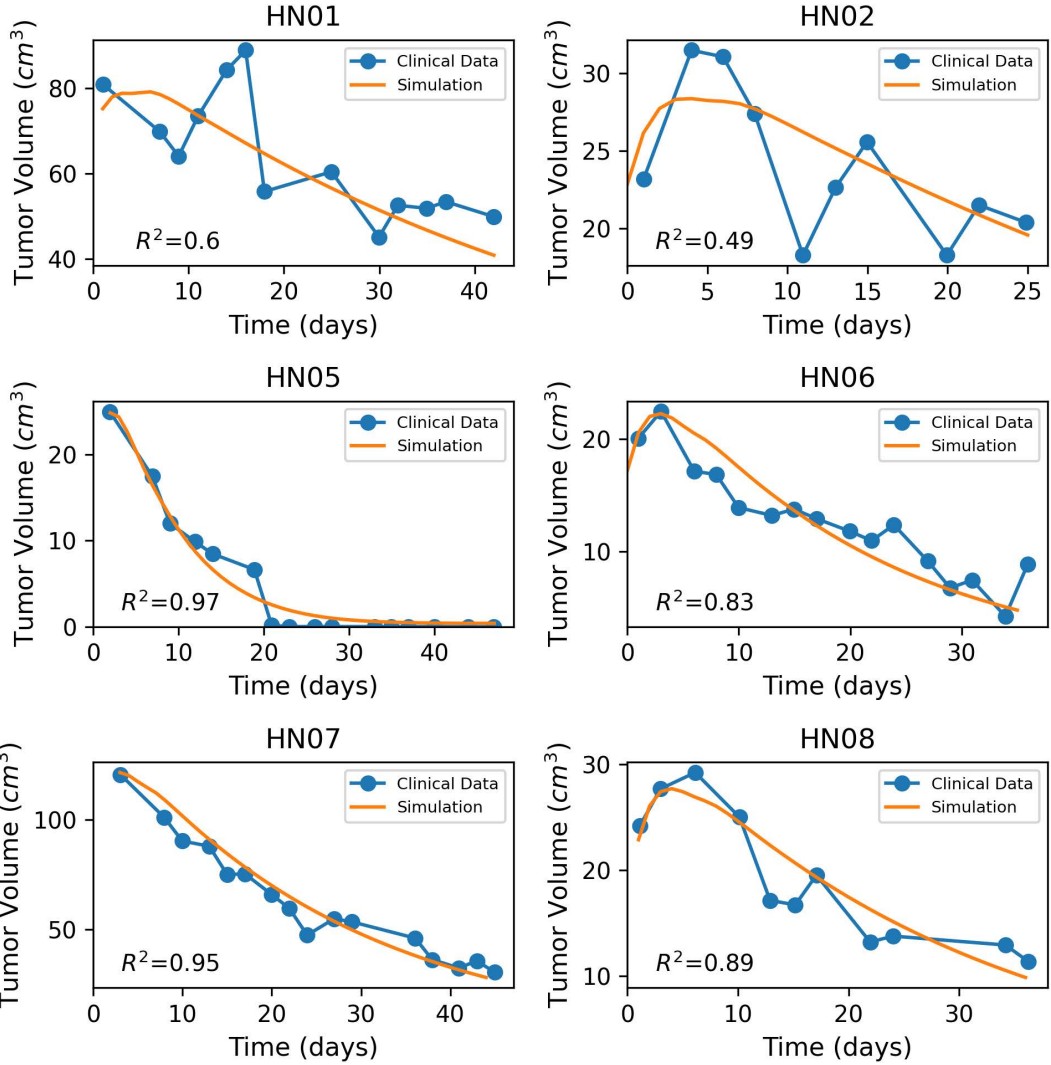

**Fig 5. Fitting of the model to clinical data of head-and-neck cancer patients [16].** Blue line with dots denotes the clinical data and the orange line the model predictions.

## Optimal treatment schedule for head and neck cancer

Simulations were performed using the parameters of patient HN15 to analyze the impact of different treatments on tumor volume under varying tumor proliferation rates. Specifically, two sets of treatment response curves were generated for the conventional and hyperfractionated treatment. The first set represents a proliferation rate that is double the baseline, while the second set corresponds to a proliferation rate that is quintuple the baseline (Fig 7). For the case where the proliferation rate is doubled, a small divergence between the two treatment response curves is observed, suggesting a modest advantage for hyperfractionated treatment. However, when the proliferation rate increases to five times the baseline, the divergence between the curves becomes significantly larger. This indicates a greater benefit of hyperfractionated over the conventional treatment schedule in managing more aggressive (highly proliferated) tumors. It demonstrates that hyperfractionation may have a more pronounced effect in improving treatment outcomes as tumor proliferation rates increase. Same conclusions are reached when parameters for other patients are considered.

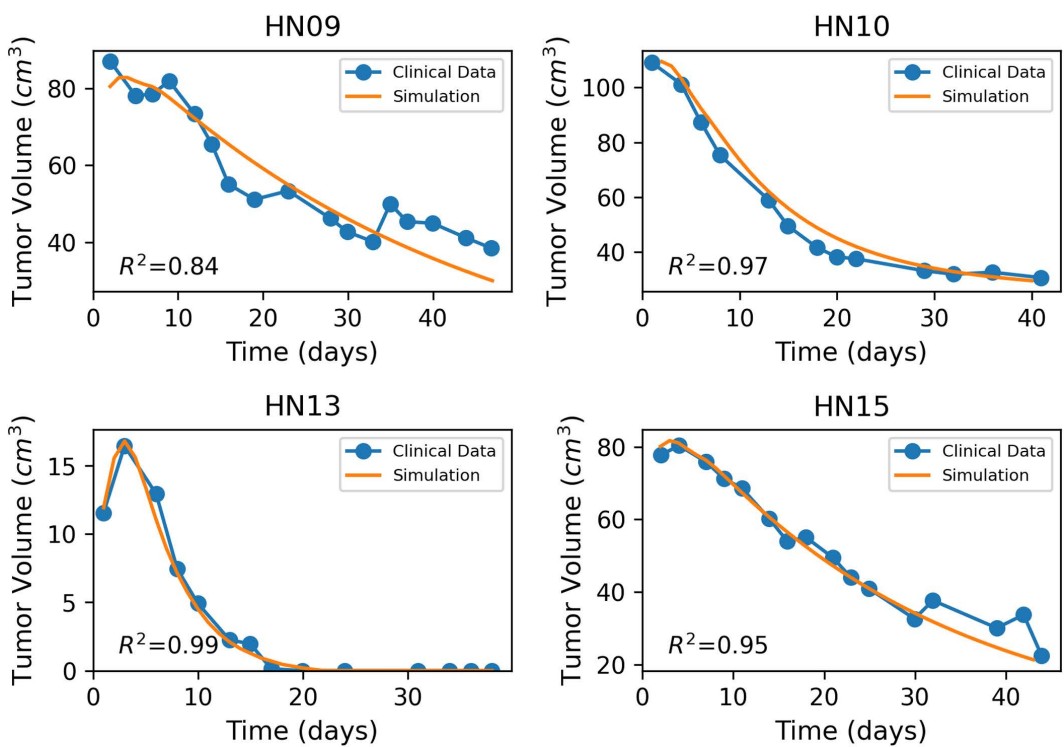

**Fig 6. Fitting of the model to clinical data of head-and-neck cancer patients [16].** Blue line with dots denotes the clinical data and the orange line the model predictions.

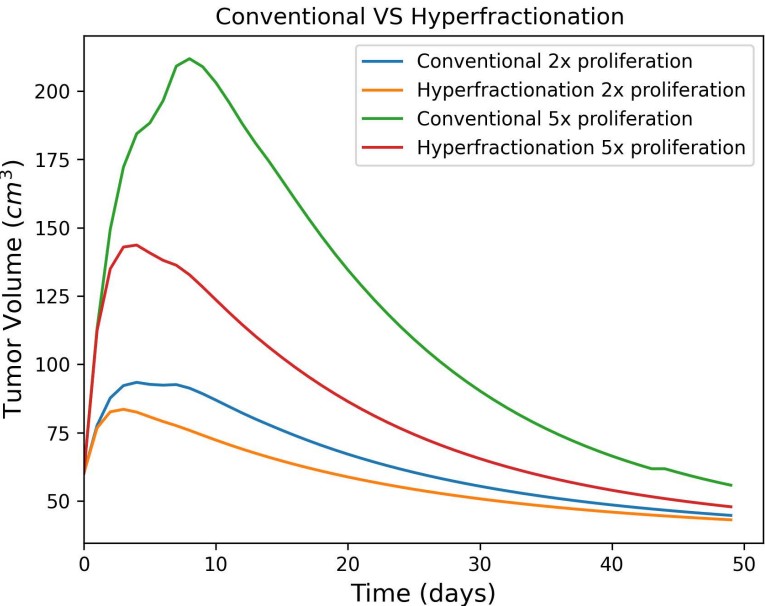

**Fig 7. The impact of different treatments on tumor volume under varying tumor proliferation rates for head and neck patients.** The results indicate a greater benefit of hyperfractionated treatment over the conventional one in managing more aggressive tumors.

## Optimal treatment schedule for Prostate cancer

Following the validation of the head and neck cases, we modified the model to simulate the growth of prostate tumors during radiotherapy. Considering that prostate cancer grows at a slower rate compared to head and neck cancer, we reduced parameter $k_M$ to one-eighth of the value used in the head and neck simulation. In addition, the clearance rate $C_r$ was set to a value equal to that observed in patients HN06 and HN10 to facilitate the analysis. It should be noted that, due to lack of available data in the literature, parameters $k_M$ and $C_r$ could not be fitted as they were in the previously mentioned cases. However, they were chosen based on our daily clinical experience treating prostate cancer patients at the hospital and to ensure that the model's predictions stayed both reasonable and physiologically relevant.

Simulations were performed for three different radiotherapy schemes, to investigate their effect on prostate tumor growth: conventionally fractionated radiotherapy (2 Gy once a day, 5 days a week, 78 Gy in total), hypo-fractionated radiotherapy (3 Gy once a day, 5 days a week, 60 Gy in total) and stereotactic body radiotherapy (SBRT) (7.25 Gy once a day, every other day, 36.25 Gy in total) [64]. The parameters of the Linear Quadratic Model used in each simulation are presented in Table 6. Subsequently, a parametric analysis regarding the influence of tumor perfusion and oxygenation, described by the functional vascular density $S_v$, on the effectiveness of radiotherapy was conducted. Specifically, the functional vascular density was modified immediately before the first treatment session, with low and high vascular densities represented by 5000 m$^{-1}$ and 25000 m$^{-1}$, respectively [37]. Lower functional vascular density results in reduced perfusion and oxygenation within the tumor, whereas higher vascular density corresponds to normal perfusion and oxygenation levels.

Fig 8 depicts the total number of surviving tumor cells as a function of time for the multiple radiotherapy schemes. The simulation results show that all radiotherapy schemes manage to eliminate all tumor cells, by the end of the treatment. As a result, tumor recurrence does not occur in any of the cases. Due to the schedule of each treatment, stereotactic body is the fastest in achieving maximum cell kill compared to the rest of the schemes, while conventionally fractionation is the slowest.

Fig 9 illustrates the effect of tumor functional vascular density on the percentage of surviving tumor cells, for three different radiosensitivity scenarios of the stereotactic body radiotherapy treatment. Starting with Fig 9A, higher functional vascular density appears to enhance treatment effectiveness by reducing the number of surviving cancer cells throughout the duration of therapy. The curve corresponding to higher vascular density consistently lies below that of lower vascular density, indicating a greater reduction in tumor cell population over time. However, by the end of the treatment period, both conditions ultimately result in a similar percentage of cancer cell kill. Fig 9B displays the percentage of surviving cancer cells during the treatment in a more radioresistant situation. Particularly, the rise in cancer cell radioresistance is demonstrated by a decrease in parameter $\alpha$ of the Linear Quadratic model. Interestingly, as radioresistance increases, changes in functional vascular density have a greater influence on the percentage of surviving cells. Specifically, the curve

**Table 6. Model parameters and units for model predictions of prostate cancer.**

| Parameters | Value | Reference |
|---|---|---|
| $\frac{\alpha}{\beta}$ (Gy) | 1.78 | [64] |
| $\alpha_{G_1}$ (Gy$^{-1}$) | 0.043 | [64] |
| $\alpha_S$ (Gy$^{-1}$) | 0.043 | [64] |
| $\alpha_{G_2}$ (Gy$^{-1}$) | 0.043 | [64] |
| $\alpha_M$ (Gy$^{-1}$) | 0.043 | [64] |
| $\alpha_{G_0}$ (Gy$^{-1}$) | 0.043 | [64] |
| $C_r$ (day$^{-1}$) | 1/70 | This work |
| $k_M$ (hour$^{-1}$) | 0.001 | This work |

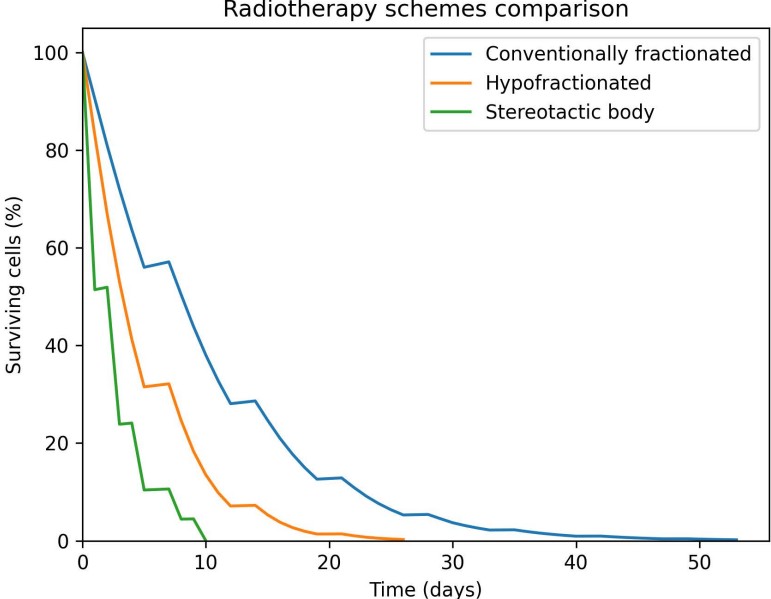

**Fig 8. Radiotherapy schemes comparison.** All radiotherapy schemes manage to eliminate all tumor cells, by the end of the treatment.

associated with higher vascular density remains below that of lower vascular density across the timeline. Ultimately, this results in a significantly lower number of surviving cancer cells by the end of the treatment. Finally, Fig 9C also represents a more radioresistant scenario compared to the one in Fig 9A. In this case, the increase in radioresistance resulted from a higher $\alpha/\beta$ ratio of the Linear Quadratic model. Similarly to the scenario of Fig 9B, an elevated functional vascular density ends up with a lower percentage of surviving cancer cells, indicating a more efficient treatment.

Subsequently, we studied the effect of functional vascular density on the percentage of surviving tumor cells under three different radiosensitivity scenarios in hypofractionated treatment. Beginning with Fig 10A, higher functional vascular density appears to improve treatment effectiveness, with consistently lower cancer cell survival throughout therapy. However, both conditions ultimately achieve a similar level of surviving cells by the end of the treatment. Fig 10B represents the percentage of surviving cancer cells during the treatment, with a reduced $\alpha$ parameter, thus making the tumor cells more radioresistant. It is illustrated that the treatment becomes more effective with higher functional vascular density, as its curve remains lower throughout the treatment and ultimately leads to fewer surviving cancer cells. Finally, Fig 10C depicts the case in which $\alpha/\beta$ ratio is increased, thus making the tumor more radioresistant compared to the one in Fig 10A. As with the other cases, a higher functional vascular density results in a lower percentage of surviving cancer cells over the course of the treatment. At the end of the therapy, the condition with higher functional vascular density shows slightly lower number of surviving cancer cells. Importantly, we observe that the same key observation noted in SBRT treatment also applies to hypofractionated treatment: as radioresistance augments, alterations in functional vascular density have a larger effect on the percentage of surviving cells.

Finally, Fig 11 presents the effect of functional vascular density on the percentage of surviving tumor cells, for three different radiosensitivity scenarios of the conventionally fractionated treatment. As it can be seen, the results are similar to those of stereotactic body radiotherapy and hypofractionated treatment. Specifically, greater functional vascular density improves treatment effectiveness by increasing the number of cancer cells killed, across the treatment timeline. In addition, as radioresistance increases, changes in functional vascular density have a greater impact on the percentage of surviving cells. This is evident in Fig 11B, where higher vascular density results in a significantly lower number of surviving cells by the end of the therapy.

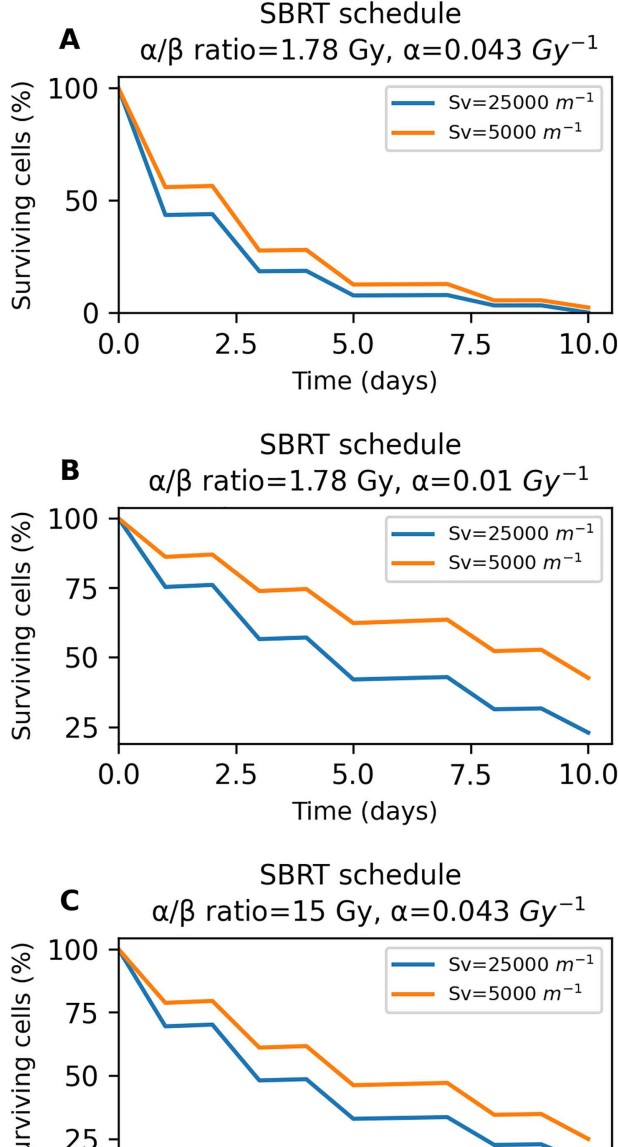

**Fig 9. Effect of functional vascular density on the percentage of surviving tumor cells, for three different radiosensitivity scenarios of the SBRT: Stereotactic Body Radiotherapy treatment.** **A** graph corresponds to the baseline scenario while **B** and **C** panels correspond to increased radioresistance. The blue and orange curves represent high and low functional vascular densities, respectively.

It should be noted that the results shown in Figs 9–11 are based on deterministic model simulations. As such, these figures are intended to highlight how the system responds to changes in vascular density, focusing on overall trends rather than statistically significant differences.

## Discussion

We present a mathematical model based on continuum mechanics principles to simulate tumor progression under radiotherapy treatment. It is the first continuous mathematical model designed to investigate various radiotherapy schedules

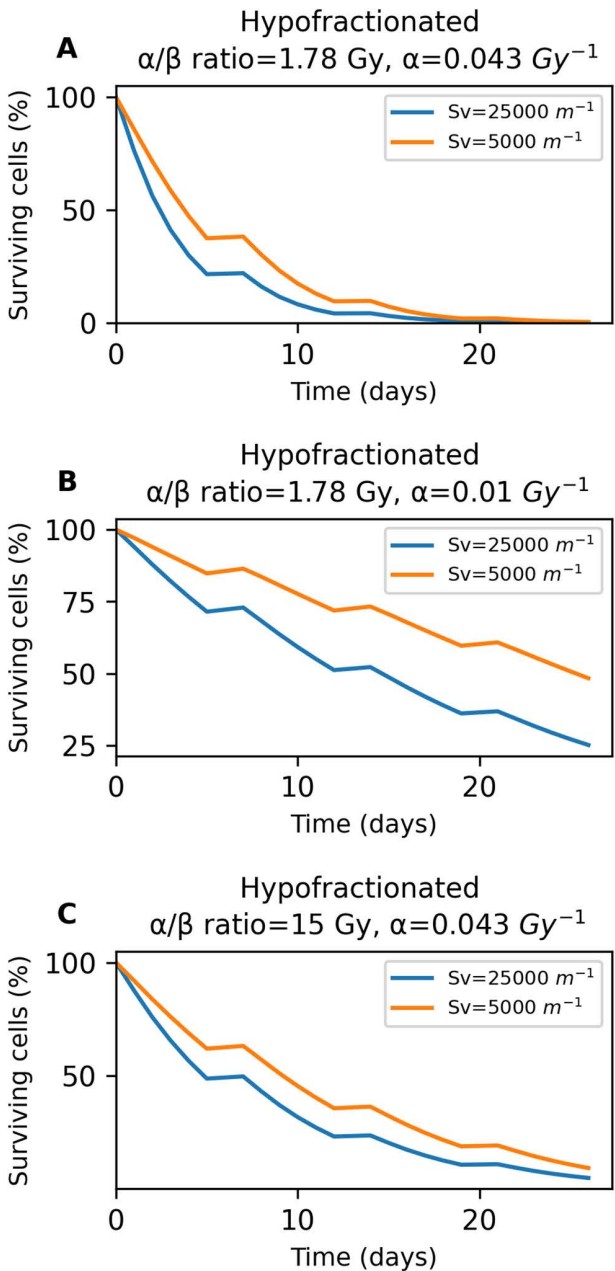

**Fig 10. Effect of functional vascular density on the percentage of surviving tumor cells, for three different radiosensitivity scenarios of the hypofractionated treatment.** **A** graph corresponds to the baseline scenario, while **B** and **C** correspond to increased radioresistance. The blue and orange curves represent high and low functional vascular densities, respectively.

while incorporating alterations in perfusion and oxygenation. Moreover, this framework is general, allowing for the integration of new insights about the tumor's radiosensitivity as they become available. Lastly, the model demonstrates its adaptability during its validation on pre-clinical as well as clinical data, on different type of cancers.

The most important finding of the study is the impact of vascular density and oxygenation on determining the effectiveness of radiation therapy. Particularly, simulations were performed for three different radiotherapy schemes, to investigate

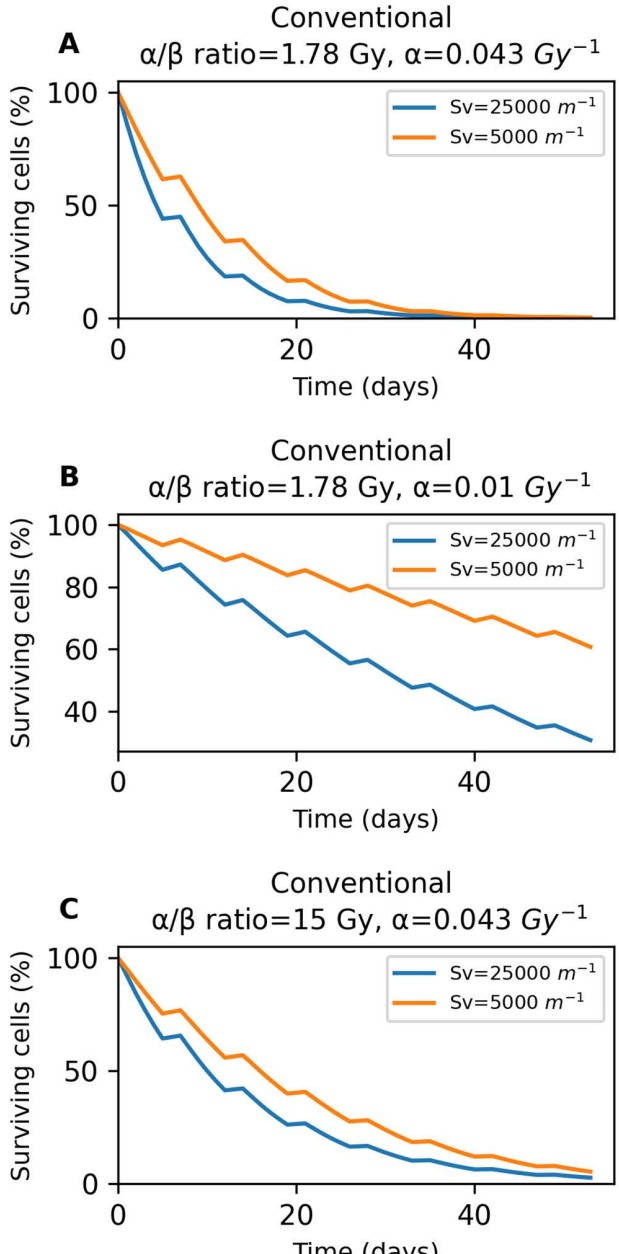

**Fig 11. Effect of functional vascular density on the percentage of surviving tumor cells, for three different radiosensitivity scenarios of the conventionally fractionated treatment.** **A** graph corresponds to the baseline scenario, while **B** and **C** correspond to increased radioresistance. The blue and orange curves represent high and low functional vascular densities, respectively.

their effect on tumor growth, focusing on values of model parameters that characterize prostate tumors. The following radiotherapy schemes have been simulated: conventionally fractionated radiotherapy (2 Gy once a day, 5 days a week, 78 Gy in total), moderate hypofractionated radiotherapy (3 Gy once a day, 5 days a week, 60 Gy in total) and stereotactic body radiotherapy (SBRT) (7.25 Gy once a day, every other day, 36.25 Gy in total). In addition, we incorporated mechanisms into our model to examine the effects of oxygen variability on tumor growth. Specifically, the value of the

functional vascular density $S_v$ is modified, resulting in altered oxygen levels within the tumor region. All three radiotherapy treatments showed similar results regarding the effect of functional vascular density on the percentage of surviving cells. Notably, greater functional vascular density improves treatment effectiveness by increasing the number of cancer cells killed during the treatment period. In addition, as radioresistance increases, changes in functional vascular density have a greater impact on the percentage of surviving cells. Specifically, increased functional vascular density during treatment of radioresistant tumors leads to fewer surviving cancer cells by the end of therapy.

Moreover, the findings of the radiotherapy schemes comparison can be quite meaningful (Fig 8). When the three radio-therapy schemes are compared to each other, all treatments manage to control the tumor (kill all tumor cells), by the end of the treatment, as seen in Fig 8. Due to the schedule of each treatment, SBRT is the fastest in achieving maximum cell kill compared to the rest of the schemes, while conventionally fractionation is the slowest. As a result, SBRT could be considered as optimal for patients with intermediate-risk or high-risk prostate cancer. The rationale behind this is the fact that SBRT requires fewer treatment sessions compared to the other methods. This approach benefits both patients, by reducing the duration of their treatment, and the radiotherapy department, allowing them to treat more patients in the same period, thus helping to shorten waiting lists. Nevertheless, the results of the simulation model should always be interpreted with caution since any expected normal tissue complications were not considered.

Several simplifying assumptions were applied in this study. It is well established that cell radiosensitivity varies significantly throughout the cell cycle. Therefore, each cell phase is modelled with different $\alpha$ and $\beta$, resulting in different death rates after exposure to radiation [18,28,45]. However, clinical studies typically aim to determine the $\alpha/\beta$ ratio for each tumor type, often overlooking the intra-cycle variation radiosensitivity. As a result, the same values for $\alpha$ and $\beta$ were used during the validation of the murine prostate model and the predictions for prostate cancer patients, owing to the limited availability of phase-specific data in the literature. Additionally, most of the parameters used were independently obtained from literature. Nevertheless, for parameters lacking available data, values were chosen to ensure that the model's predictions remained both reasonable and physiologically relevant. An additional assumption is the use of simplified geometry and the consideration of isotropic cancer cell proliferation. Future steps involve incorporating realistic geometries that account for structural inhomogeneities. Therefore, the tumor will be modeled to proliferate anisotropically, with distinct growth stretch ratios $\lambda_g$ assigned to each spatial direction [17,65]. This approach enables a more accurate representation of the inherently heterogeneous growth patterns observed in solid tumors. Furthermore, the material properties of the tumour can be derived from a cancer patient, along with data on the radiosensitivity of the surrounding tissue. As a result, the model will become highly personalizable. Additionally, the model does not account for immune responses. Future work should incorporate mechanisms which consider how irradiation can either suppress or stimulate immune response [66]. Specifically, on one hand radiation triggers various forms of tumor cell death, leading to the release of pro-inflammatory cytokines, chemokines, tumor antigens and other danger-associated signals. This mechanism can increase the immunogenicity of the tumor. Therefore, combining local radiotherapy with immunotherapy could improve the effectiveness of cancer treatment [67]. On the other hand, lymphocytes (T cells, B cells and NK) are some of the most radiosensitive cells, thus they can be depleted during radiotherapy. As a result, the combination of radiotherapy and immunotherapy does not consistently produce a synergistic effect [67]. Taking into account mechanisms related to anti-tumor immune responses will likely affect the predicted by the model cell killing rate. Excluding these effects may lead to either an over- or underestimation of treatment efficacy, depending on tumor's immune profile. Incorporating such mechanisms could improve the accuracy of predictions, particularly in cases where immune dynamics play a significant role in patient outcomes. Another limitation of our study is the absence of normal tissue complication probability (NTCP) calculations. In the context of prostate SBRT, a recent randomized controlled trial reported an increased risk of cumulative genitourinary toxicities of grade 2 or higher (26.9% vs. 18.3% p < 0.001) [14]. In the context of head and neck cancer, a large multi-institutional comparison found that severe acute toxicity of

grade 4 or higher was significantly less frequent with Stereotactic Ablative Body Radiotherapy compared to Intensity-Modulated Radiation Therapy (0.5% vs. 5.1%) [68]. Future studies should incorporate NTCP assessments to provide a more comprehensive understanding of the therapeutic ratio and facilitate potential translation into clinical practice. An additional limitation of our study is the application of the model to prostate cancer in the absence of validation with data from prostate cancer patients. In general, treatment efficacy in prostate cancer is evaluated indirectly by monitoring prostate-specific antigen (PSA) levels during follow-ups, rather than by directly measuring changes in tumor volume during treatment. Therefore, there is a lack of clinical tumor growth data to compare model predictions with. To model the response of prostate tumors, we tailored tumor-specific parameters ($C_r$ and $k_M$) using values from published data and from our clinical experience to ensure the model remains accurate and applicable to prostate cancer cases. Given this limitation, it is important to emphasize that the prostate cancer modeling presented here should be regarded as exploratory. While this approach allows for reasonable simulations aligned with daily clinical observations, it does not replace the need for rigorous clinical validation. As such, the results for prostate cancer should be interpreted with caution, and future work should prioritize parameter estimation based on direct clinical measurements or PSA-derived tumor kinetics, as data becomes available.

While in vivo validation would further strengthen the conclusions, the current study is focused on computational modeling and simulation-based insights. To support the biological plausibility of our findings, we reference relevant studies that have demonstrated associations between decreased perfusion, tumor hypoxia, and reduced radiotherapy efficacy [55,57,69,70].

The key finding of this study is that vascular density and tissue oxygenation play a critical role in determining the effectiveness of radiation therapy. Consequently, strategies aimed at enhancing oxygen availability—such as vascular and stroma normalization—can improve therapeutic outcomes. One approach to achieve vascular normalization is through the use of anti-angiogenic agents, which can restore vessel hyper-permeability and restructure the abnormal vascular network [53,71,72]. Several clinical studies have demonstrated that anti-angiogenic therapy can induce normalization of tumor vasculature [71,73,74]. In addition, stroma normalization can be facilitated through with the use of mechanotherapeutics [53,75–78]. Mechanotherapeutics aim to normalize tumor stroma by reducing tumor stiffness and intratumoral mechanical forces. This can be achieved either by directly depleting extracellular matrix components or by reprogramming cancer-associated fibroblasts, ultimately leading to vessel decompression. Based on pertinent clinical studies, even one week of pre-treatment with a mechanotherapeutic, it is sufficient to normalize the tumor stroma of pancreatic tumors and sarcomas [79,80]. Anti-angiogenic treatment to normalize the tumor vasculature is time- and dose-dependent and not easy to control because elongated treatment or high dose can cause excessive vessel pruning and thus, reduced perfusion [72]. Therefore, close monitoring of vessel functionality during anti-angiogenic treatment with medical imaging methods is required.

While this study focuses on stereotactic body radiotherapy, hypofractionated and conventionally fractionated treatments, emerging radiotherapy methods are rapidly advancing the field. Techniques like Adaptive Radiotherapy (ART) and Surface-Guided Radiotherapy (SGRT) enhance precision by adjusting to daily anatomical changes and tracking patient positioning in real time. Alongside AI-driven imaging, MR-LINACs, particle therapy and emerging approaches such as FLASH radiotherapy and radioimmunotherapy, these innovations offer more personalized, targeted treatments that improve tumor control while minimizing damage to healthy tissue [81–83].

The presented methodology and results demonstrate that the developed model can provide useful insights into the effectiveness of various radiotherapy treatment protocols for prostate as well as head and neck cancers. It addresses the importance of oxygenation and perfusion in enhancing the effectiveness of the treatment. Finally, the model can generalize well on other types of cancer and could serve as a tool for studying cancer biology and aiding radiotherapy decisions.

## Supporting information

**S1 Text. Sensitivity analysis on the transition rate $k_M$, the clearance rate $C_r$ and the parameter $\alpha$ of the Linear Quadratic model.**
(PDF)

**S1 Table. The logarithm of total variance in tumor volume for the parameters of the sensitivity analysis (log(*Tot*. *Var*.)>1).**
(PDF)

**S1 Fig. Bland–Altman plot of patient HN01 comparing modeled-predicted tumor volumes with experimental measurements.** Each dot represents the difference versus the mean of the two measurements for a single time point.
(TIF)

**S2 Fig. Bland–Altman plot of patient HN02 comparing modeled-predicted tumor volumes with experimental measurements.** Each dot represents the difference versus the mean of the two measurements for a single time point.
(TIF)

**S3 Fig. Bland–Altman plot of patient HN05 comparing modeled-predicted tumor volumes with experimental measurements.** Each dot represents the difference versus the mean of the two measurements for a single time point.
(TIF)

**S4 Fig. Bland–Altman plot of patient HN06 comparing modeled-predicted tumor volumes with experimental measurements.** Each dot represents the difference versus the mean of the two measurements for a single time point.
(TIF)

**S5 Fig. Bland–Altman plot of patient HN07 comparing modeled-predicted tumor volumes with experimental measurements.** Each dot represents the difference versus the mean of the two measurements for a single time point.
(TIF)

**S6 Fig. Bland–Altman plot of patient HN08 comparing modeled-predicted tumor volumes with experimental measurements.** Each dot represents the difference versus the mean of the two measurements for a single time point.
(TIF)

**S7 Fig. Bland–Altman plot of patient HN09 comparing modeled-predicted tumor volumes with experimental measurements.** Each dot represents the difference versus the mean of the two measurements for a single time point.
(TIF)

**S8 Fig. Bland–Altman plot of patient HN10 comparing modeled-predicted tumor volumes with experimental measurements.** Each dot represents the difference versus the mean of the two measurements for a single time point.
(TIF)

**S9 Fig. Bland–Altman plot of patient HN13 comparing modeled-predicted tumor volumes with experimental measurements.** Each dot represents the difference versus the mean of the two measurements for a single time point.
(TIF)

**S10 Fig. Bland–Altman plot of patient HN15 comparing modeled-predicted tumor volumes with experimental measurements.** Each dot represents the difference versus the mean of the two measurements for a single time point.
(TIF)

**S11 Fig. Sensitivity analysis on the parameter $\alpha$ of the Linear Quadratic model, to evaluate its influence on the model outputs.** The parameter varied at five levels, 1.1, 1.05, 1, 0.95 and 0.9 times its baseline value as given in Table 2.
(TIF)

**S12 Fig. Sensitivity analysis on the clearance rate $C_r$, to evaluate its influence on the model outputs.** The parameter varied at five levels, 1.1, 1.05, 1, 0.95 and 0.9 times its baseline value as given in Table 2.
(TIF)

**S13 Fig. Sensitivity analysis on the transition rate $k_M$, to evaluate its influence on the model outputs.** The parameter varied at five levels, 1.1, 1.05, 1, 0.95 and 0.9 times its baseline value as given in Table 2.
(TIF)

## Author contributions

**Conceptualization:** Yiannis Roussakis, Constantinos Zamboglou, Triantafyllos Stylianopoulos.

**Data curation:** Kyprianos Dimou.

**Formal analysis:** Kyprianos Dimou.

**Funding acquisition:** Constantinos Zamboglou, Triantafyllos Stylianopoulos.

**Investigation:** Kyprianos Dimou, Yiannis Roussakis, Constantinos Zamboglou, Triantafyllos Stylianopoulos.

**Methodology:** Kyprianos Dimou, Yiannis Roussakis, Constantinos Zamboglou, Triantafyllos Stylianopoulos.

**Project administration:** Yiannis Roussakis, Constantinos Zamboglou, Triantafyllos Stylianopoulos.

**Resources:** Kyprianos Dimou, Yiannis Roussakis, Constantinos Zamboglou, Triantafyllos Stylianopoulos.

**Software:** Kyprianos Dimou.

**Supervision:** Yiannis Roussakis, Constantinos Zamboglou, Triantafyllos Stylianopoulos.

**Validation:** Kyprianos Dimou.

**Visualization:** Kyprianos Dimou.

**Writing – original draft:** Kyprianos Dimou.

**Writing – review & editing:** Kyprianos Dimou, Yiannis Roussakis, Constantinos Zamboglou, Triantafyllos Stylianopoulos.

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
