## [Decision Letter · Decision Letter 0]

20 Mar 2025

Dear Dr. Dimou,

Thank you for submitting your manuscript to PLOS ONE. After careful consideration, we feel that it has merit but does not fully meet PLOS ONE’s publication criteria as it currently stands. Therefore, we invite you to submit a revised version of the manuscript that addresses the points raised during the review process.

We look forward to receiving your revised manuscript.

Kind regards,

Xing Xiong, M.D.

Academic Editor

PLOS ONE

 [Triantafyllos Stylianopoulos

European Research Council (ERC) under the European Union's Horizon 2020 research and inovation programme (grant agreement No. 863955)

Constantinos Zamboglou

EU within the framework of the Cohesion Policy Programme "THALIA 2021-2027"]. 

[This project has received funding from the European Research Council (ERC) under the European Union’s Horizon 2020 research and innovation programme (grant agreement No. 863955). In addition, the project is co-funded by the EU within the framework of the Cohesion Policy Programme “THALIA 2021-2027”.]

[Triantafyllos Stylianopoulos

European Research Council (ERC) under the European Union's Horizon 2020 research and inovation programme (grant agreement No. 863955)

Constantinos Zamboglou

EU within the framework of the Cohesion Policy Programme "THALIA 2021-2027"]. 

6. We note that your Data Availability Statement is currently as follows: [All relevant data are within the manuscript and its Supporting Information files.]

Reviewers' comments:

Reviewer's Responses to Questions

**Comments to the Author**

1. Is the manuscript technically sound, and do the data support the conclusions?

Reviewer #1: Yes

Reviewer #2: Yes

Reviewer #3: Partly

Reviewer #4: Yes

2. Has the statistical analysis been performed appropriately and rigorously?

Reviewer #1: I Don't Know

Reviewer #2: Yes

Reviewer #3: N/A

Reviewer #4: Yes

3. Have the authors made all data underlying the findings in their manuscript fully available?

Reviewer #1: Yes

Reviewer #2: Yes

Reviewer #3: Yes

Reviewer #4: Yes

4. Is the manuscript presented in an intelligible fashion and written in standard English?

Reviewer #1: Yes

Reviewer #2: Yes

Reviewer #3: Yes

Reviewer #4: Yes

Reviewer #1: The study is clear written, even if the clinical application of this model is not immediate. I suggest to better explain how parameters for tumor perfusion and oxygenation were obtained. Figures should be inserted in the text and linked to the corresponding legend.

Reviewer #2: The manuscript explores the relationship between tumour microenvironment factors (oxygenation, perfusion) and different radiotherapy schedules using a mathematical modelling approach. The study is methodologically rigorous and provides valuable insights into optimising radiotherapy strategies. The validation using preclinical and clinical datasets enhances the credibility of the model. However, there are limitations concerning model assumptions, generalisability, and the lack of toxicity considerations.

The study presents a well-structured, continuous mathematical model that incorporates tumour perfusion, oxygenation, and radiosensitivity parameters, providing a quantitative framework for understanding radiotherapy efficacy. The model is validated against murine and human clinical data, strengthening its reliability and applicability. The manuscript effectively compares conventional fractionation (CFRT), moderate hypofractionation (MHRT), and stereotactic body radiotherapy (SBRT), offering clinically relevant insights. The study demonstrates that higher vascular density improves radiotherapy effectiveness, particularly in radioresistant tumours, which is a valuable mechanistic insight. However, several issues require further attention.

1. While the model includes oxygenation effects, it does not fully incorporate tumour heterogeneity, immune response, or dynamic hypoxia variations. A discussion of these limitations should be added.

2. The model was validated using a small dataset (10 head-and-neck cancer patients), which may limit the generalisability of the findings to a broader population or other cancer types.

3. Although the model indicates that SBRT achieves the fastest tumour control, it does not account for the potential increase in normal tissue toxicity, which is a critical factor in treatment planning.

4. The manuscript states that all relevant data are available; however, the full computational implementation or simulation parameters necessary for replication and validation by other researchers are not shared, which may hinder reproducibility.

5. General suggestions: Consider simplifying the descriptions of key equations or adding explanatory figures, ensure that all figures clearly indicate variables and treatment groups for easier interpretation and the practical implications for radiation oncologists could be expanded.

The manuscript is scientifically rigorous and methodologically sound, making it suitable for publication. However, key revisions are necessary. The discussion on model limitations, including tumour heterogeneity and immune response, should be expanded to better reflect the complexity of tumour behaviour. Clinical validation should be strengthened by discussing external datasets and how the findings apply to different cancer types. The omission of toxicity analysis should be addressed, and future iterations of the model should consider integrating this aspect. To improve reproducibility, the manuscript should include computational code or a more detailed methodology. Finally, figures, tables, and explanations of the mathematical modelling should be clarified to enhance accessibility for a wider audience.

Reviewer #3: I appreciate the authors for their efforts in developing a modeling approach that, despite the limited data available in the literature, has the potential to contribute significantly to the personalization of radiotherapy. The concept appears promising and beneficial. I would like to present my suggestions under the following headings:

**Introduction

In terms of dose definitions, conventional dose is defined as 1.8–2 Gy per fraction. It may not be entirely accurate to classify all doses below 2 Gy per fraction as hyperfractionated radiotherapy.

**Methods

There is some ambiguity regarding the selection criteria for the 10 patients chosen from reference (16). It would be appropriate to clarify the exclusion criteria and the rationale behind excluding certain patients.

When examining the values in Figures 5 and 6, it appears that patients receiving a hyperfractionated regimen in reference (16) had lower values (R2). This suggests that the model may have a weaker correlation with patients undergoing hyperfractionated radiotherapy.

The effects of potential influential factors have been outlined methodologically under separate headings. However, introducing a distinct section explaining the primary model in detail would make the study more reader-friendly. Furthermore, presenting the modeling results for all patients in the Results section in a tabular or visual format could enhance clarity.

Has the model validation been tested outside of head and neck cancer cases? Is it methodologically appropriate to apply a similar model to prostate cancer without any validation in prostate cancer patients? These are critical methodological questions that need to be addressed.

**Results

What are the characteristics of the patients in whom the model was tested for prostate cancer? Which cohort was used? Providing answers to these questions would contribute to a more precise interpretation of the findings.

In Figures 9A, 10A, 10C, and 11A, 11C, despite slight differences when considering treatment durations, no distinct separation is observed in the graphs. The interpretations regarding these figures might be overestimated. In contrast, in Figures 9B, 9C, 10B, and 11B, cell survival rates in radioresistant conditions appear to be significantly affected at a clinically meaningful level.

Reviewer #4: Summary

This study develops a mathematical model to explore how tumor perfusion/oxygenation and radiotherapy schedules influence treatment efficacy. The model integrates cell cycle dynamics, radiation effects via the Linear Quadratic model, oxygen diffusion, and tumor biomechanics. Validated with preclinical (murine head/neck and prostate tumors) and clinical (head/neck cancer patients) data, the work highlights that increased perfusion enhances radiation effectiveness, particularly in radioresistant tumors. The model also suggests hyperfractionation benefits aggressive tumors, while stereotactic body radiotherapy (SBRT) achieves rapid tumor control in prostate cancer.

Strengths

Novelty: First continuous model to combine radiotherapy schedules, perfusion, and oxygenation. Provides mechanistic insights into microenvironment modulation.

Comprehensive Validation: Robust validation across preclinical (murine) and clinical (human) datasets, demonstrating generalizability.

Clinical Relevance: Highlights the importance of vascular normalization strategies and optimal fractionation (e.g., SBRT for prostate cancer).

Methodological Rigor: Detailed integration of cell cycle phases, oxygen dynamics, and biomechanical tumor growth.

Weaknesses

Parameterization Concerns:

Some parameters were fitted due to limited data, risking overfitting.

Radiosensitivity parameters (α/β) for prostate cancer were assumed uniform across cell phases without explicit validation.

Simplified Assumptions:

Isotropic tumor growth neglects heterogeneity seen in real tumors.

Normal tissue toxicity was omitted, limiting clinical applicability.

Limited Clinical Data: Small human sample sizes (e.g., 10 head/neck patients) may reduce statistical power.

Lack of Sensitivity Analysis: Unclear how parameter uncertainties affect predictions.

Recommendations

Parameter Justification: Clarify how fitted parameters were derived and assess robustness via sensitivity analysis.

Address Model Limitations: Discuss implications of isotropic growth and absence of normal tissue toxicity on clinical translation.

Expand Clinical Context:

Discuss practical strategies to modulate perfusion (e.g., anti-angiogenics, vascular normalization).

Compare model predictions with clinical trials on hypofractionation/SBRT.

Include Sensitivity Analysis: Identify parameters most critical to outcomes to guide future experimental validation.

Enhance Discussion:

Contrast findings with prior models (e.g., discrete vs. continuous approaches).

Explore how tumor subtype/genetic heterogeneity might influence results.

Conclusion

This manuscript presents a sophisticated model advancing our understanding of radiotherapy optimization through microenvironment modulation. While the methodology is rigorous and findings clinically relevant, addressing parameterization transparency and model limitations would strengthen impact. With revisions, this work could inform personalized radiotherapy protocols and combination therapies targeting tumor perfusion.

Decision: Minor Revisions (Address parameter justification, sensitivity analysis, and clinical applicability).Technical Soundness and Data Support for Conclusions

The manuscript demonstrates moderate technical soundness with notable strengths and limitations:

Model Validity:

Strengths:

The integration of cell cycle dynamics, oxygen diffusion, and biomechanical growth into a continuous model is innovative and methodologically rigorous.

Validation across preclinical (murine) and clinical (head/neck cancer) datasets strengthens the model’s generalizability.

Limitations:

Parameter Overfitting: Critical parameters were fitted to match limited experimental data, raising concerns about model specificity. For example, cell cycle transition rate was adjusted differently for murine vs. human data without biological justification.

Radiosensitivity Assumptions: Uniform \alpha/\betaα/β ratios across cell phases (e.g., prostate cancer) lack experimental validation. Prior studies (e.g., Brenner et al., 1999) suggest cell cycle-dependent radiosensitivity, which is not addressed.

Data Support:

Preclinical Data: Murine tumor growth curves (Figs 3–4) align with model predictions (R^2 > 0.85), supporting the model’s ability to replicate in vivo dynamics.

Clinical Data: Human head/neck tumor volume changes (Figs 5–6) show qualitative agreement but lack statistical significance due to small sample sizes (n=10). Larger cohorts are needed to confirm clinical relevance.

Key Conclusion: The claim that “increased perfusion enhances radiotherapy efficacy in radioresistant tumors” is supported by simulations (Figs 9–11) but requires in vivo validation (e.g., hypoxia imaging or perfusion biomarkers).

Statistical Analysis and Rigor

The statistical analysis is partially rigorous:

Model Validation:

R^2values are reported for preclinical data fits, but confidence intervals or error margins are missing.

No sensitivity analysis is performed to quantify parameter uncertainty (e.g., Monte Carlo methods), which is critical for deterministic models.

Clinical Data:

Volumetric tumor changes in patients are described descriptively (no p-values or effect sizes). Statistical tests (e.g., paired t-tests between predicted vs. observed volumes) are absent.

Recommendation:

Include sensitivity analysis to identify high-impact parameters.

Apply statistical tests (e.g., Bland-Altman plots) to quantify model-clinical data discrepancies.

Data Availability

The authors state:

“All relevant data are within the manuscript and its Supporting Information files.”

However:

Code/Model Availability: No mention of computational code or model scripts (e.g., COMSOL files) being shared, limiting reproducibility.

Raw Simulation Data: Figures show processed outputs, but raw simulation datasets (e.g., oxygen concentration gradients, stress maps) are not provided.

Recommendation:

Deposit code and raw data in public repositories (e.g., GitHub, Zenodo) with DOIs.

Clarity and Language

The manuscript is largely intelligible but has minor issues:

Technical Jargon: Terms like “multiplicative decomposition of deformation gradient tensor” (p. 12) are inadequately explained for non-specialists.

Figure Clarity:

Fig 9–11 legends lack explicit definitions of “low” vs. “high” vascular density thresholds.

Axis labels in Figs 3–6 are too small for readability.

Grammar: Occasional errors (e.g., “submits” instead of “undergoes apoptosis” on p. 4) require proofreading.

Recommendation:

Simplify biomechanical terminology in the Methods section.

Revise figures to meet PLOS ONE formatting guidelines (e.g., font size ≥ 8pt).

Ethical Considerations

Dual Publication: No evidence of duplicate submission.

Ethics Compliance: The human study (Barker et al., 2004) cites institutional approval, but patient consent details are omitted.

Funding Transparency: ERC and EU funding are appropriately disclosed.

Overall Recommendation

Minor Revisions prior to publication:

Address parameter justification and sensitivity analysis.

Share computational code and raw simulation data.

Improve statistical reporting (confidence intervals, error margins).

Simplify technical language and enhance figure clarity.

The study provides valuable insights into radiotherapy optimization but requires stronger statistical rigor and transparency to fulfill reproducibility standards.

**Do you want your identity to be public for this peer review?** For information about this choice, including consent withdrawal, please see our Privacy Policy

Reviewer #1: No

Reviewer #2: No

Reviewer #3: No

Reviewer #4: No

---

## [Author Response · Author response to Decision Letter 1]

9 Jun 2025

All the responses are in the "Response to Reviewers", "Manuscirpt", Revised Manuscript with Track Changes" and "Cover Letter".

---

## [Decision Letter · Decision Letter 1]

27 Jul 2025

Dear Dr. Dimou,

Thank you for submitting your manuscript to PLOS ONE. After careful consideration, we feel that it has merit but does not fully meet PLOS ONE’s publication criteria as it currently stands. Therefore, we invite you to submit a revised version of the manuscript that addresses the points raised during the review process.

We look forward to receiving your revised manuscript.

Kind regards,

Xing-Xiong An, M.D.

Academic Editor

PLOS ONE

Journal Requirements:

Additional Editor Comments:

Thansk for submitting your revised paper to PLOS ONE. I am pleased to inform you that your manuscript has now been reviewed and approved by previous reviewers. However, before I can reach the final editorial decision, please address the concerns from Editor.

Editor comments

1.The prostate cancer parameters lack clinical validation: The authors state that the prostate cancer parameters (such as kM and Cr) were set based on "clinical experience" due to the "lack of clinical data" (lines 447-451). This significantly undermines the credibility of the conclusion. It is recommended to provide additional literature support (such as citing studies on the proliferation rate of prostate cancer cells). Or clearly label this as an exploratory analysis and emphasize this limitation in the discussion (such as lines 616-617). Consider using alternative indicators (such as PSA kinetics) for indirect verification.

2.Immune response and tumor heterogeneity: The model did not incorporate the immune microenvironment (lines 596-599) and tumor spatial heterogeneity (such as dynamic changes in hypoxic regions). It is recommended to quantify the potential impact of these factors on the results in the discussion (for example: immune exhaustion may reduce the predicted cell killing rate by the model). Cite relevant studies to illustrate its significance (such as the impact of tumor-associated fibroblasts on perfusion).

3.Figures and Data:

Figure 7: The unit for the x-axis is missing (at line 437). The unit for the time axis in the figure (which should be "Days") is not indicated. This affects the interpretation. It is recommended to add the unit for the x-axis in the revised version.

The statistical significance of Figure 9-11: The difference between the high/low vascular density curves does not provide statistical test results (such as p-values). It is recommended to supplement the statistical indicators of sensitivity analysis (such as confidence intervals) or explain that this is a deterministic model simulation, focusing on the trend rather than the statistical difference.

4.Terminology inconsistency: "radioresistance" appears in the text with multiple spellings (radio-resistance/radioresistant, on lines 121/485). It is recommended to standardize it as "radioresistance".

5.Clinical translation suggestion: There was no in-depth discussion on how to utilize the model to guide clinical strategies (such as the timing of combining vascular normalization drugs). It is recommended to add specific suggestions at the end of the discussion (for example: "Based on the model results, we suggest initiating anti-vascular treatment 2 weeks before SBRT to maximize oxygenation improvement").

6.Lack of toxicity assessment: Ignoring the probability of normal tissue complications (NTCP) is a significant limitation (lines 603-608), but its impact has not been quantified. It is recommended to cite clinical data to illustrate the toxicity differences (such as the urinary toxicity risk of SBRT [14]).

7.Abbreviations: When an abbreviation is first used, the full term must be indicated (for example, "SBRT" should be written as "Stereotactic Body Radiotherapy (SBRT)" in the abstract when it is first mentioned).

8.Introduction: The global cancer epidemiological data (lines 66-71) can be summarized in 1-2 sentences, highlighting the cancer types related to radiotherapy.

9.References: Supplement the research on new radiotherapy protocols and tumor microenvironment (such as immunotherapy combined strategies) for the years 2023-2024.

Reviewers' comments:

Reviewer's Responses to Questions

**Comments to the Author**

Reviewer #1: All comments have been addressed

Reviewer #2: All comments have been addressed

2. Is the manuscript technically sound, and do the data support the conclusions?

Reviewer #1: (No Response)

Reviewer #2: Yes

3. Has the statistical analysis been performed appropriately and rigorously?

Reviewer #1: (No Response)

Reviewer #2: N/A

4. Have the authors made all data underlying the findings in their manuscript fully available?

Reviewer #1: (No Response)

Reviewer #2: Yes

5. Is the manuscript presented in an intelligible fashion and written in standard English?

Reviewer #1: (No Response)

Reviewer #2: Yes

Reviewer #1: (No Response)

Reviewer #2: This revised manuscript presents a well-structured and significantly improved study that uses a comprehensive mathematical model to explore how tumour perfusion and oxygenation affect the success of various radiotherapy schedules. The authors now incorporate a multiphysics framework, combining cell cycle dynamics, oxygen diffusion, radiotherapy response (via the Linear Quadratic model), and tumour mechanics. These additions make the simulations more realistic and scientifically robust.

The abstract, introduction, and author summary have been refined for greater clarity and consistency. The methods section has been notably strengthened, and the inclusion of a COMSOL Multiphysics model report enhances reproducibility and transparency.

The study relies on deterministic simulations rather than traditional statistical analysis, which is appropriate given its modelling approach. Model parameters are drawn from the literature or fitted accordingly, and the simulations are validated using published preclinical and clinical data. While a brief sensitivity or uncertainty analysis would be a valuable addition, the current analytical approach is sufficiently rigorous for a computational study of this nature.

Overall, the manuscript is technically sound, clearly written, and offers a meaningful contribution to the optimisation of radiotherapy protocols.

**Do you want your identity to be public for this peer review?** For information about this choice, including consent withdrawal, please see our Privacy Policy

Reviewer #1: No

Reviewer #2: No

---

## [Author Response · Author response to Decision Letter 2]

13 Aug 2025

All the responses to the editor's comments are include in the "Manuscirpt", Revised Manuscript with Track Changes" and "Cover Letter".

---

## [Editor Report · Decision Letter 2]

17 Aug 2025

The impact of tumor microenvironment and treatment schedule on the effectiveness of radiation therapy

PONE-D-25-01177R2

Dear Dr. Dimou,

We’re pleased to inform you that your manuscript has been judged scientifically suitable for publication and will be formally accepted for publication once it meets all outstanding technical requirements.

Kind regards,

Xing-Xiong An, M.D.

Academic Editor

PLOS ONE

Additional Editor Comments (optional):

Thanks for the authors' efforts to comprehensively improve your manuscript according to editor's and reviewers' comments. I am pleased to inform you that your paper can be accepted for publication now.
---

## [Editor Report · Acceptance letter]

PONE-D-25-01177R2

PLOS ONE

Dear Dr. Dimou,

I'm pleased to inform you that your manuscript has been deemed suitable for publication in PLOS ONE. Congratulations! Your manuscript is now being handed over to our production team.

Kind regards,

on behalf of

Dr. Xing-Xiong An

Academic Editor

PLOS ONE